# Survival and Death Causes in Thyroid Cancer in Taiwan: A Nationwide Case–Control Cohort Study

**DOI:** 10.3390/cancers13163955

**Published:** 2021-08-05

**Authors:** Yu-Ling Lu, Shu-Fu Lin, Ming-Hsien Wu, Yi-Yin Lee, Pai-Wei Lee, Shang-Hung Chang, Yu-Tung Huang

**Affiliations:** 1Department of Internal Medicine, New Taipei Municipal TuCheng Hospital, New Taipei City 230617, Taiwan; smartlynn18@cgmh.org.tw (Y.-L.L.); mmg@cgmh.org.tw (S.-F.L.); b9502013@cgmh.org.tw (M.-H.W.); winnielee@cgmh.org.tw (Y.-Y.L.); 2Endocrinology & Metabolism Division, Department of Internal Medicine, Chang Gung Memorial Hospital, Linkou Main Branch, Taoyuan City 333423, Taiwan; 3School of Medicine, College of Medicine, Chang Gung University, Taoyuan City 333323, Taiwan; afen.chang@gmail.com; 4Center for Big Data Analytics and Statistics, Chang Gung Memorial Hospital, Linkou Main Branch, Taoyuan City 333423, Taiwan; paiwei@cgmh.org.tw; 5Cardiovascular Division, Department of Internal Medicine, Chang Gung Memorial Hospital, Linkou Main Branch, Taoyuan City 333423, Taiwan

**Keywords:** thyroid cancer, causes of death, survival, Taiwan

## Abstract

**Simple Summary:**

This study aims to evaluate overall survival and the risk of cause-specific mortality of thyroid cancer patients. Thyroid cancer patients were obtained from the universal health insurance claims from Taiwan between 2001 and 2017. We compared these patients with control subjects matched for age, gender, and baseline conditions to assess the risk of mortality. Of the 30,778 patients with thyroid cancer, the overall mortality rate was 1.29% and the leading causes of death were thyroid cancer (31.2%), other cancers (29.9%), and cardiovascular disease (CVD) mortality (12.3%), respectively. We found patients with thyroid cancer had excellent overall survival and lower CVD mortality risk.

**Abstract:**

The incidence of thyroid cancer has increased substantially worldwide. However, the overall mortality risk and actual causes of death in thyroid cancer patients have not been extensively evaluated. In this study, patients with thyroid cancer diagnosed between 2001 and 2017 were analyzed from Taiwan’s National Health Insurance Research Database. We compared these patients with control subjects matched for age, gender, history of cardiovascular disease (CVD), hyperlipidemia, diabetes mellitus, hypertension, and occupation to assess the risk of overall mortality and cause-specific mortality. Finally, our cohort comprised 30,778 patients with thyroid cancer. Three hundred and ninety-eight deaths (1.29%) occurred during a median follow-up of 60.0 months (range: 30.3 to 117.6 months). The primary cause of death was thyroid cancer mortality (31.2%), followed by other malignancy-related mortality (29.9%) and CVD mortality (12.3%). The overall mortality risk was similar between the thyroid cancer and control groups (unadjusted hazard ratio (HR): 0.98; 95% confidence interval (CI): 0.88–1.10); the adjusted HR was 1.07 (95% CI: 0.95–1.20) after multivariate adjustment for age, gender, history of CVD, hyperlipidemia, diabetes mellitus, hypertension, and occupation. The risk of other malignancy-related mortality was comparable between two groups. CVD mortality risk was lower in the thyroid cancer group, with an unadjusted HR of 0.51 (95% CI: 0.38–0.69) and adjusted HR of 0.56 (95% CI: 0.42–0.76). In conclusion, patients with thyroid cancer had excellent overall survival. Thyroid cancer-specific mortality was the leading cause of death, highlighting the importance of thyroid cancer management. Thyroid cancer patients had lower CVD mortality risk than the general population.

## 1. Introduction

Thyroid cancer is the most common malignancy of the endocrine system, and the incidence of thyroid cancer has rapidly increased over the past four decades worldwide [1,2,3,4]. Despite this increase in incidence, mortality from thyroid cancer remains low and stable. In the USA, the death rate rose by just over 0.5% annually between 2009 and 2018, with an overall 5-year survival rate of 98% [2]. Similarly in Taiwan, thyroid cancer has demonstrated a 2.2-fold increase in incidence between 1997 and 2012, with steady 5-year survival rates between 1997 (90.2%) and 2010 (92.4%) [3].

The increasing number of thyroid cancer patients who generally have favorable long-term survival rates raises the question if these subjects have worse overall mortality compared with the general population. Other questions are the actual causes of death in these patients and relative risks of cause-specific mortality compared with the general population. A nationwide study was conducted to evaluate the overall mortality in thyroid cancer patients in Korea [5]. Excellent overall survival (98.5% survival rate) was observed during a median of a 48-month follow-up in 4082 thyroid cancer patients. Of note, the mortality risk was comparable between thyroid cancer patients and the general population. Unfortunately, this study has limitations in the sample size and duration of follow-up. Additional studies to confirm the findings of this study that similar mortality risk existed between thyroid cancer patients and the general population are mandatory.

Prior studies have evaluated actual causes of death in thyroid cancer patients [5,6,7]. The results of these studies demonstrated the leading causes of death in thyroid cancer patients were thyroid cancer-related (32.8–69.7%), other malignancies (11.6–31.1%), and cardiovascular disease (CVD; 6.25–13.2%). Regarding CVD, it is one of the major causes of mortality in thyroid cancer patients. Some studies have been conducted to compare the risk of cardiovascular mortality in patients with thyroid cancer versus the general population [5,8,9]. A study revealed increased risks of CVD mortality in thyroid cancer patients as compared to the control population [8], while another two studies did not [5,9]. These studies had important limitations, including small sample sizes of thyroid cancer patients and CVD history that was not adjusted between thyroid cancer patients and the general population.

In this study, the primary aim was to evaluate overall survival of thyroid cancer patients compared with the general population. The secondary aim was to assess the actual causes of death and the risk of cause-specific mortality in thyroid cancer patients versus the general population.

## 2. Materials and Methods

### 2.1. Source of Data

This retrospective study analyzed data from the Taiwan National Health Insurance (NHI) Research Database (NHIRD). NHIRD was initiated in 1997 by the Bureau of National Health Insurance to enhance research on health care in Taiwan, which covers about 99% of Taiwan residents (approximately 23.7 million people) [10,11]. NHIRD comprises detailed information on patient demographic profiles (date of birth, gender, and occupation), clinical information (diagnoses, examination, procedures, and surgeries related to inpatient and ambulatory care), and prescribed medications. Personal identification is anonymized but linkable between the NHIRD and other government databases, such as the Birth Registry and the Death Registry [11]. The Taiwan Death Registry dataset was also included in this study and the information on the date and causes of death for each deceased subject was evaluated [12]. This dataset is considered to be complete and accurate because all death certificates must be completed via physicians by law in Taiwan [12,13]. All the study datasets and data analyses were obtained and approved by the Health and Welfare Data Center established by Taiwan’s Ministry of Health and Welfare.

### 2.2. Ascertainment of Subjects with Thyroid Cancer

Subjects who were the holders of thyroid cancer catastrophic illness cards registered in the Catastrophic Illness Registry database, a subdataset of NHIRD, were identified and were considered to have thyroid cancer in this study. A thyroid cancer catastrophic illness card is issued for copayment and coinsurance exemption under NHI after a medical claim review of pathological findings and approval by experts of the NHI administration [14].

### 2.3. Study Cohort

We conducted a retrospective cohort study to compare overall survival and cause-specify mortality between thyroid cancer patients and matched controls. Data in the Taiwan NHIRD dataset and Death Registry dataset between 1 January 2001 and 31 December 2017 were retrospectively analyzed. Patients who had thyroid cancer catastrophic illness cards were included in the thyroid cancer cohort. Individuals who met the following criteria were excluded: patients who had received thyroidectomy 6 months prior to a thyroid cancer diagnosis, who did not undergo thyroidectomy after a diagnosis of thyroid cancer, who had a history of radioiodine or thyroxine treatment, who had a prior diagnosis of other malignancies, and for whom the demographic data was missing. For choosing subjects for the control group, the individuals who met the following criteria were excluded: patients who had a prior diagnosis of thyroid disease, who had a history of thyroid surgery, who had undergone radioiodine or thyroxine treatment, who died of thyroid cancer, and for whom the demographic data was missing, and had a diagnosis of other malignancies before the index date. The date of the application for thyroid cancer catastrophic illness cards was identified as the index date. We used the propensity score matching (PSM) method to balance covariates between the study groups [15]. Two-stage PSM was applied regarding the considerable number (*n* = 29,083,313) of beneficiaries in the NHIRD during this study period. Birth year was used for initial matching (1:20; *n* = 32,711 vs. *n* = 625,549), followed by using age, gender, comorbidities (including ischemic heart disease, ischemic stroke, hemorrhagic stroke, hyperlipidemia, diabetes mellitus, and hypertension), and occupation for further matching (1:3; *n* = 30,778 vs. *n* = 92,334). The diagnosis of these diseases was according to International Classification of Diseases, ninth and tenth revision (ICD-9 and ICD-10). ICD-9 was used between 2001 and 2015 and ICD-10 was implemented since 2016 [16]. The list of these ICD codes is shown in Appendix A.

### 2.4. Covariates

Covariates that could potentially affect survival, including patient age, gender, history of cardiovascular disease (ischemic heart disease, ischemic stroke, and hemorrhagic stroke), hyperlipidemia, diabetes mellitus, hypertension, and occupation were identified and included in our analyses [17,18]. The occupation was categorized into 5 groups (dependents of the insured individuals; civil servants, teachers, military personnel, and veterans; non-manual workers and professionals; manual workers; others) according to NHI coverage classification [4,19].

### 2.5. Study Outcomes

Data on the date and causes of death were acquired from the Death Registry dataset. The underlying cause of death was utilized in the mortality analyses. The primary end point of this study was the relative risk of all-cause mortality between thyroid cancer patients and the general population. The secondary end points were cause-specific mortality risk between these two groups. The causes of death analyzed in this study were the leading causes of mortality in Taiwan, including malignancy, cardiovascular disease, diabetes mellitus, infectious disease, injury/trauma, renal disease, digestive disease, lower respiratory disease, and anemia/malnutrition/age-related debility [20].

### 2.6. Statistical Analyses

PSM technique was used to reduce the bias of confounding variables that could lead to mortality. The quality of PSM was assessed using the absolute standardized mean difference (ASMD) between two groups, where a value <0.1 was considered to a have negligible difference [21]. Continuous variables with non-normal distribution were presented as the median and interquartile range (IQR). Dichotomous variables were presented as percentages. The *χ*^2^ test was employed to compare the cause-specific mortality between thyroid cancer patients and matched controls. The hazard ratios (HR) and 95% confidence intervals (CI) were computed using the Cox proportional hazards model. The all-cause mortality and cause-specific mortality were estimated using Kaplan–Meier survival analyses and the log-rank test. A two-sided *p* < 0.05 was considered statistically significant. All analyses were performed with SAS software version 9.4 (SAS Institute Inc., Cary, NC, USA).

### 2.7. Ethics Statement

This study was conducted in accordance with the ethical guidelines of the 1975 Declaration of Helsinki and was approved by the institutional review board of the Chang Gung Medical Foundation (No. 1711270055) and the Health and Welfare Data Science Center, Ministry of Health and Welfare, Taiwan (No. H107030). Informed consent was waived due to all personal information being anonymized. 

## 3. Results

### 3.1. Patient Characteristics of the Study Population

The flowchart depicting the identification of subjects in the thyroid cancer group and the control group is shown in Figure 1. A total of 29,083,313 individuals were identified, including 53,028 subjects with catastrophic illness cards of thyroid cancer and 29,030,285 subjects without this card in the NHIRD dataset between 2001 and 2017. Individuals were excluded if they met the exclusion criteria as described above. After initial matching for birth year (1:20), followed by secondary matching for age, gender, histories of ischemic heart disease, ischemic stroke, hemorrhagic stroke, hyperlipidemia, diabetes mellitus, hypertension, and occupation (1:3), there were 30,778 and 92,334 individuals in the thyroid cancer group and the control group, respectively.

The thyroid cancer group consisted of 6,581 males and 24,197 females, with a median age of 47 years (IQR: 37–56 years) at the time of thyroid cancer diagnosis (Table 1). Patients with a history of ischemic heart disease, ischemic stroke, and hemorrhagic stroke appeared in 1.7% (537/30,778), 0.4% (114/30,778), and 0.2% (66/30,778) of the thyroid cancer group, respectively. Incidences of hyperlipidemia, diabetes mellitus, and hypertension occurred in 1.7% (518/30,778), 5.1% (1565/30,778), and 13.8% (4234/30,778) of thyroid cancer patients, respectively. The proportions of subjects having ischemic heart disease, hemorrhagic stroke, hyperlipidemia, and diabetes mellitus were well balanced between the thyroid cancer group and the control group. However, the thyroid cancer group had lower rates of ischemic stroke (0.4% vs. 1.6%) and higher rates of hypertension (13.8% vs. 7.0%) compared with the control group. Distributions of five occupation categories were closely matched between two groups. Most thyroid cancer patients underwent total or subtotal thyroidectomy (56.8% and 4.9%, respectively). Approximately a third of thyroid cancer patients received radioiodine treatment (10,146/30,778; 33.0%), with a median cumulative dose at 120 mCi (IQR: 100–200 mCi). The median follow-up time was 68.0 months (IQR: 30.3–117.6 months) in the thyroid cancer group.

### 3.2. Causes of Death and Overall Mortality Risk

A total of 398 deaths occurred in the thyroid cancer group and 1318 deaths in the control group during the study period. Table 2 summarizes the specific causes of death. Thyroid cancer-related death (124/398, 31.2%) was the most common cause of mortality in the thyroid cancer group, followed by other malignancies (119/398, 29.9%) and cardiovascular disease (49/398, 12.3%). The numbers of mortality attributed to diabetes mellitus, infectious disease, injury/trauma, renal disease, digestive disease, lower respiratory disease, and anemia/malnutrition/age-related debility were limited (*n* ≤ 20) in the thyroid cancer group. In the control group, the most frequent cause of death was cardiovascular disease (323/92,334, 24.5%), followed by malignancies other than thyroid cancer (313/92,334, 23.8%).

All-cause mortality was similar between the thyroid cancer group and the control group (1.29% vs. 1.43%), with an unadjusted HR of 0.98 (95% CI, 0.88–1.10), which was at 1.01 (95% CI, 0.90–1.13) after adjustment for age and sex, and remained insignificant at 1.07 (95% CI, 0.95–1.20) after adjustment for age, gender, ischemic heart disease, ischemic stroke, hemorrhagic stroke, hyperlipidemia, diabetes mellitus, hypertension, and occupation (Table 3). Other malignancy-related death was also comparable between the thyroid cancer group and the control group (29.9% vs. 23.8%), with an unadjusted HR of 1.17 (95% CI, 0.94–1.44), which was at 1.19 (95% CI, 0.96–1.47) after adjustment for age and sex, and remained insignificant at 1.21 (95% CI, 0.98–1.49) after adjustment for age, gender, ischemic heart disease, ischemic stroke, hemorrhagic stroke, hyperlipidemia, diabetes mellitus, hypertension, and occupation. Considering cardiovascular disease-related mortality, the thyroid cancer group had a lower mortality rate than that of the control group (12.3% vs. 24.5%), with an unadjusted HR of 0.51 (95% CI, 0.38–0.69), which was at 0.53 (95% CI, 0.39–0.71) after adjustment for age and sex, and stayed significantly at 0.56 (95% CI, 0.42–0.76) after adjustment for age, gender, ischemic heart disease, ischemic stroke, hemorrhagic stroke, hyperlipidemia, diabetes mellitus, hypertension, and occupation.

The Kaplan–Meier survival curves for all-cause mortality, other malignancy-related mortality, and CVD-related mortality are illustrated in Figure 2. No significant differences were observed between the thyroid cancer group and the control group in all-cause mortality (*p* = 0.783; Figure 2A) and other malignancy-related mortality (*p* = 0.153; Figure 2B). However, the thyroid cancer group had a lower risk of having CVD mortality (*p* < 0.001; Figure 2C).

## 4. Discussion

We performed a nationwide case–control cohort study to assess the overall survival and cause-specific mortality risk in thyroid cancer patients. Our data demonstrate that thyroid cancer patients had similar overall survival compared with the general population. In addition, thyroid cancer patients had a comparable risk of other malignancy-related mortality and a lower risk of cardiovascular mortality. These data are important regarding the increasing incidence of thyroid cancer. Taking these issues of mortality into consideration is mandatory during management and follow-up of thyroid cancer patients.

Thyroid cancer-related death is the most common cause (31.2%) of mortality in this study. Our findings are consistent with prior reports showing thyroid cancer-related death is the most frequent reason contributing to death in this population, ranging from 69.7% in 1970–1985, 65.0% in 1986–2000, and 32.8% in 2002–2014 [5,6,7]. Of note, the proportion of thyroid cancer patients who died of thyroid cancer decreased from 69.7% to 31.2% during the past four decades [5,6,7]. The reasons contributing to the lower percentage of thyroid cancer patient deaths due to thyroid cancer are unclear and need to be clarified.

We found that one of the leading causes of death in thyroid cancer patients is other malignancies (29.9%). The result is consistent with prior data demonstrating other malignancies accounted for approximately one third of deaths in thyroid cancer patients (between 31.1% and 35%) [6,7]. In addition, our data showed that thyroid cancer patients did not carry higher mortality risk due to other malignancies compared to the general population, although 33% of patients underwent radioiodine therapy with a median accumulated dose of 120 mCi in this study. It has been reported that an increase in cumulative radioiodine dose confers higher risk for second primary malignancy in thyroid cancer patients, especially with accumulative doses over 150 mCi [22].

CVD is the third leading cause of mortality in thyroid cancer patients in this study. However, thyroid cancer patients had lower risk of mortality due to CVD compared with the general population. The reasons accounting for this phenomenon have to be clarified. The potential explanations can be early recognition of CVD and risk management during the follow-up after the diagnosis of thyroid cancer. The finding of thyroid cancer patients having lower CVD mortality is in line with a study including 901 differentiated thyroid cancer patients with a median follow-up of 18.8 years [9], but is in contrast to a study consisting of 524 differentiated thyroid cancer patients with a median follow-up of 8.5 years showing thyroid cancer patient had higher CVD mortality [8].

This study exhibits strengths. The major strengths include the population-based design and sufficient sample size. The number of thyroid cancer patients analyzed in this study was much more than that in prior studies (30,778 vs. 366–4082) [5,6,7,8,9], which allowed us to study the mortality outcomes in a disease with favorable long-term survival.

This study has some limitations. First, the design of this observational study prevented us from drawing any conclusions about potential causal relations between decreased CVD mortality and the associated factors. Patients with a history of ischemic stroke were less prevalent in the thyroid cancer group at baseline compared with the controls that might contribute to the lower incidence of CVD mortality in the thyroid cancer group. However, after adjustment for this condition, the risk of CVD mortality remained lower. Additional studies are needed to confirm our results of lower cardiovascular disease-related mortality in the thyroid cancer group. Second, we did not include pathological subtypes and tumor stage that can affect a thyroid cancer prognosis. The vast majority of thyroid cancer patients in Taiwan were differentiated thyroid cancer (papillary and follicular thyroid cancer), accounting for 91–96% of the whole thyroid cancer patient population [4]. Patients with differentiated thyroid cancer usually have favorable outcomes, with a 10-year disease-specific survival of 96% [23]. Regarding the fact that more than 90% of thyroid cancers were differentiated thyroid cancer in Taiwan, our data were likely derived mainly from patients with differentiated thyroid cancer. Third, we did not evaluate the correlation between thyroid stimulating hormone (TSH) suppression therapy and mortality risk due to the limitation of the dataset. A study consisting of 524 differentiated thyroid cancer patients with a median follow-up of 8.5 years reported a reverse association between TSH levels and cardiovascular mortality rate [8]. Fourth, we did not have information on cardiovascular risk factors, such as smoking, drinking, and body mass index, which are associated with cardiovascular mortality [24,25]. Fifth, we excluded patients who did not undergo thyroid surgery. Therefore, our results could not be extrapolated to these subjects. A study revealed a worse 5-year survival rate in thyroid cancer patients who did not undergo surgical treatment compared with those who did (56.8% vs. 96.7%). Advanced age, later tumor stage, and more frequent distant metastases occurred in the nonsurgical group in that study [26]. Sixth, we did not evaluate the association between risk of second primary cancer mortality and radioiodine therapy. A prior report has revealed an increased risk of secondary primary cancer in thyroid cancer patients, especially in whom received cumulative radioiodine doses over 150 mCi [22]. It is reasonable to speculate the existence of an association between risk of second primary cancer mortality and radioiodine therapy. Additional studies are needed to evaluate this possibility. Seventh, our findings were based on the analysis of data from Taiwan NHIRD dataset. Further analyses of other national cohorts are needed to address survival and death causes in thyroid cancer in other countries. Eighth, we did not specifically evaluate the survival and death causes of each histologic type of thyroid cancer compared to that of the general population. Additional studies are needed to address the limitations of this study.

## 5. Conclusions

In conclusion, by studying a nationwide database, we found patients with thyroid cancer had similar overall survival as that of the general population. The most common cause of death was thyroid cancer-specific mortality, indicating the importance of thyroid cancer management. The risk of other malignancy-related mortality, the second leading cause of death in thyroid cancer patients, was not meaningfully different between thyroid cancer subjects and the general population. Thyroid cancer patients had lower cardiovascular mortality risk.

## Figures and Tables

**Figure 1 cancers-13-03955-f001:**
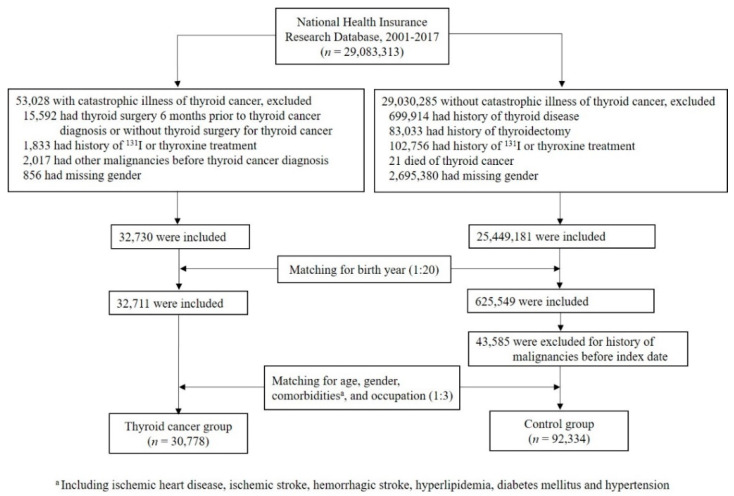
Flow chart of study populations.

**Figure 2 cancers-13-03955-f002:**
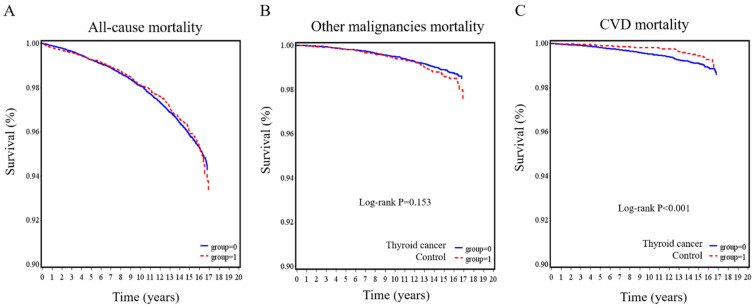
Kaplan–Meier survival curves for thyroid cancer patients and controls. (**A**) all-cause, (**B**) other malignancies, and (**C**) cardiovascular disease mortality in the thyroid cancer group compared with the control group.

**Table 1 cancers-13-03955-t001:** Characteristics of the study subjects.

	Thyroid Cancer Group	Control Group	ASMD
	(*n* = 30,778)	(*n* = 92,334)	
Age, median (years, IQR)	47 (37–56)	47 (37–57)	0.009
Gender			
Male, *n* (%)	6581 (21.4)	19,569 (21.2)	0.005
Female, *n* (%)	24,197 (78.6)	72,765 (78.8)	0.005
History of cardiovascular disease	686 (2.2)	4124 (4.5)	0.125
Ischemic heart disease, *n* (%)	537 (1.7)	2413 (2.6)	0.060
Ischemic stroke, *n* (%)	114 (0.4)	1447 (1.6)	0.123
Hemorrhagic stroke, *n* (%)	66 (0.2)	690 (0.8)	0.078
Hyperlipidemia, *n* (%)	518 (1.7)	1371 (1.5)	0.016
Diabetes mellitus, *n* (%)	1565 (5.1)	6690 (7.3)	0.090
Hypertension, *n* (%)	4234 (13.8)	6465 (7.0)	0.223
Occupation, *n* (%)			
Category 1	16,575 (53.9)	49,068 (53.1)	0.014
Category 2	6237 (20.3)	19,540 (21.2)	0.022
Category 3	3585 (11.7)	10,874 (11.8)	0.004
Category 4	241 (0.8)	691 (0.8)	0.004
Category 5	4140 (13.5)	12,161 (13.2)	0.008
Thyroid surgery, *n* (%)			
Lobectomy	11,793 (38.3)		
Subtotal thyroidectomy	1498 (4.9)		
Total thyroidectomy	17,487 (56.8)		
Levothyroxine, *n* (%)	17,285 (56.2)		
Levothyroxine median dose (µg/day, IQR)	128.6 (100–150)		
RAI treatment, *n* (%)	10,146 (33.0)		
RAI median cumulative dose (mCi, IQR)	120 (100–200)		
Follow-up, median (months, IQR)	68.0 (30.3–117.6)	68.9 (31.0–119.1)	0.015

**Table 2 cancers-13-03955-t002:** Incidence of all-cause and cause-specific mortality in the thyroid cancer group and the control group.

	Thyroid Cancer Group (*n* = 30,778)	Control Group (*n* = 92,334)	*p* Value
	*n* (%)	*n* (%)	
All-cause mortality	398	1318	0.082
Thyroid cancer	124 (31.2)	0 (0)	<0.001
Other malignancies	119 (29.9)	313 (23.8)	0.227
Cardiovascular disease	49 (12.3)	323 (24.5)	<0.001
Diabetes mellitus	7 (1.8)	157 (11.9)	<0.001
Infectious disease	17 (4.3)	100 (7.6)	0.009
Injury/trauma	20 (5.0)	78 (6.0)	0.292
Renal disease	10 (2.5)	73 (5.5)	0.006
Digestive disease	3 (0.8)	40 (3.0)	0.006
Lower respiratory disease	6 (1.5)	26 (2.0)	0.413
Anemia, malnutrition, and age-related debility	14 (3.5)	43 (3.3)	0.936
Others	29 (7.3)	165 (12.5)	0.001

**Table 3 cancers-13-03955-t003:** Comparison of all-cause, other malignancies, and cardiovascular disease mortality between the thyroid cancer group and the controls.

Clinical Events	Mortality, *n* (%)	Unadjusted		Adjusted ^a^		Adjusted ^b^	
		HR (95% CI)	*p* Value	HR (95% CI)	*p* Value	HR (95% CI)	*p* Value
All-cause mortality							
Control (*n* = 92,334)	1318 (1.43)	1 (Reference)		1 (Reference)		1 (Reference)	
TC (*n* = 30,778)	398 (1.29)	0.98 (0.88–1.10)	0.784	1.01 (0.90–1.13)	0.850	1.07 (0.95–1.20)	0.257
Other malignancies							
Control (*n* = 92,334)	313 (0.34)	1 (Reference)		1 (Reference)		1 (Reference)	
TC (*n* = 30,778)	119 (0.39)	1.17 (0.94–1.44)	0.154	1.19 (0.96–1.47)	0.112	1.21 (0.98–1.49)	0.081
CVD mortality							
Control (*n* = 92,334)	323 (0.35)	1 (Reference)		1 (Reference)		1 (Reference)	
TC (*n* = 30,778)	49 (0.16)	0.51 (0.38–0.69)	<0.001	0.53 (0.39–0.71)	<0.001	0.56 (0.42–0.76)	<0.001

Adjusted ^a^ for age and gender. Adjusted ^b^ for age, gender, ischemic heart disease, ischemic stroke, hemorrhagic stroke, hyperlipidemia, diabetes mellitus, hypertension, and occupation.

## Data Availability

To ensure the participants’ privacy, the NHIRD and Death Registry database must be analyzed at the Health and Welfare Data Science Center, Ministry of Health and Welfare, Taiwan only.

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
