# Peer review of "Survival and Death Causes in Thyroid Cancer in Taiwan: A Nationwide Case–Control Cohort Study"

_cancers, 2021, doi:10.3390/cancers13163955_

Round 1
Reviewer 1 Report
The authors have examined the overall risk of death and cause specific mortality in thyroid cancer patients compared to a matching case control cohort. The authors have been aided by a national health system that assigns a unique identifier for patients enrolled in a Nation wide Insurance Health research Database. Of the 30,778 patients with thyroid cancer, overall mortality rate was 1.29% and the leading cause of death was thyroid cancer (31.2%), other cancers (29.9%) and CVD mortality (12.3%), respectively.They found patients with thyroid cancer had excellent overall survival and lower CVD mortality risk.
Comments
The paper is scientifically sound and well written. Limitations and strengths have been acknowledged.
It would be good for the authors to expand on whether patients with thyroid cancer who died from other cancers suffered from a second cancer related to RAI treatment.This would be a significant finding from this data which the current literature is lacking.
Author Response
We appreciate the reviewer for thoughtful review of our draft and the helpful comment. We did not evaluate the association between risk of second primary cancer mortality and radioiodine therapy in this study. A prior report has revealed an increased risk of secondary primary cancer in thyroid cancer patients, especially in whom received cumulative radioiodine doses over 150 mCi It is reasonable to speculate the existence of an association between risk of second primary cancer mortality and radioiodine therapy. Regarding the majority of patients (67%) did not receive radioiodine therapy in this study. Additional studies are needed to evaluate this possibility. We have addressed this point in revised draft (lines 299-305, 393-394).

Reviewer 2 Report
First of all, I am grateful for the privilege of having read the study submitted. The cohort design was rigorously designed and conducted with an appropriate use of statistical techniques, in particular the Two-stage PSM. The set of covariates used makes a suitable matching and adjustment of the analyzes performed. The conclusions add important information to the current state of knowledge about thyroid cancer mortality. In particular, the lower mortality risk in the thyroid cancer group and the general population (HR = 0.56) is of interest. My hope is that the research team will investigate the reasons for this interesting behaviour. I point out a few mistakes despite not being a native English speaker:
line 70: "compare" instead of "compared";
line 146: "assessed" instead of "assess";
line 244: "into" instead of "in to";
An important methodological point that deserves an explanation is the assessment of the "proportional hazards assumption".
Author Response
First of all, I am grateful for the privilege of having read the study submitted. The cohort design was rigorously designed and conducted with an appropriate use of statistical techniques, in particular the Two-stage PSM. The set of covariates used makes a suitable matching and adjustment of the analyzes performed. The conclusions add important information to the current state of knowledge about thyroid cancer mortality. In particular, the lower mortality risk in the thyroid cancer group and the general population (HR = 0.56) is of interest. My hope is that the research team will investigate the reasons for this interesting behaviour.
Response:
We appreciate the reviewer’s comments. We found thyroid cancer group had lower risk of cardiovascular disease-related mortality, with an adjustment hazard ratio of 0.56 (95% CI, 0.42–0.76) after multivariate adjustment, including age, gender, ischemic heart disease, ischemic stroke, hemorrhagic stroke, hyperlipidemia, diabetes mellitus, hypertension and occupation. However, due to the inherent limitations of NHIRD dataset, including the lack of laboratory data, body mass index, and smoking status, we were unable to analyze these covariant factors associated with cardiovascular disease-related mortality in this study. Additional studies are needed to confirm our results of lower cardiovascular disease-related mortality in thyroid cancer group. We have addressed this point in revised drat (lines 280-282).
I point out a few mistakes despite not being a native English speaker:
line 70: "compare" instead of "compared";
line 146: "assessed" instead of "assess";
line 244: "into" instead of "in to";
Response:
Many thanks, we have corrected these mistakes in revised draft (lines 70, 146, 244).
An important methodological point that deserves an explanation is the assessment of the "proportional hazards assumption".
Response:
Thanks for reminding the proportional hazards assumption. We had rechecked the Cox proportional model by the time interaction term. The p-value for all-cause mortality, other malignancies mortality, and cardiovascular disease mortality were 0.08, 0.164, and 0.677, respectively. Although these values were not very good, but acceptable for pass the proportional hazards assumption.

Reviewer 3 Report
The article "Survival and death causes in thyroid cancer in Taiwan: a nationwide case-control cohort study" by Yu-Ling Lu analyzes if thyroid cancer patients have worse overall mortality compared to the general population. The authors provide an accurate study on a large sample of the population. As such the article represents an interesting retrospective analysis evaluating overall survival and the risk of cause-specific mortality of thyroid cancer patients.
Some of the weaknesses of the article are due to the fact that, even though the developed statistical analysis seem to be more robust than previous studies thanks to the wide size of thyroid cancer patients, they remain indications which can only allow statistical considerations as the authors themselves evidence. It could of course be argued that this is good enough to reach meaningful conclusions in this specific case.
Another possible criticism could be that cardiovascular risk factors as well as the correlation between mortality risk and TSH suppression are not properly analyzed, as the authors underline in the discussion.
Anyway, the major critical aspect of the study is due to the fact that the authors do not differentiate various hystotypes of thyroid cancers which present different prognosis and use methods that exclude neoplasms with poor prognosis as anaplastic carcinomas and more severe cases in general.
It is a shareable choice to consider only well-differentiated carcinomas, but it should be stated in the methods section and it should be specified which type. Moreover, it should be examined also the outcome of these different neoplasms and their mortality rate.
Author Response
Some of the weaknesses of the article are due to the fact that, even though the developed statistical analysis seem to be more robust than previous studies thanks to the wide size of thyroid cancer patients, they remain indications which can only allow statistical considerations as the authors themselves evidence. It could of course be argued that this is good enough to reach meaningful conclusions in this specific case.
Response:
We appreciate this reviewer’s valuable comments. We acknowledge our findings were based on the analysis of data from a single nation. Data from other nationwide cohorts are needed to confirm our results. We have corrected these mistakes in revised draft (lines 305-307).
Another possible criticism could be that cardiovascular risk factors as well as the correlation between mortality risk and TSH suppression are not properly analyzed, as the authors underline in the discussion.
Response:
We found thyroid cancer group had a lower risk of cardiovascular disease-related mortality, with an adjustment hazard ratio of 0.56 (95% CI, 0.42–0.76) after multivariate adjustment, including age, gender, ischemic heart disease, ischemic stroke, hemorrhagic stroke, hyperlipidemia, diabetes mellitus, hypertension and occupation. However, due to the inherent limitations of NHIRD dataset, including the lack of laboratory data, body mass index, and smoking status, we were unable to analyze these covariant factors associated with cardiovascular disease-related mortality in this study. Additional studies are needed to confirm our results of lower cardiovascular disease-related mortality in thyroid cancer group. We have addressed this point in revised drat (lines 280-282).
Anyway, the major critical aspect of the study is due to the fact that the authors do not differentiate various hystotypes of thyroid cancers which present different prognosis and use methods that exclude neoplasms with poor prognosis as anaplastic carcinomas and more severe cases in general.
Response:
The aim of this study was to evaluate the survival and death causes of thyroid cancer compared to that of the general population in Taiwan. The survival and death causes of each histologic type of thyroid cancer compared to that of the general population need additional studies. We have addressed this point in revised draft (lines 307-310).
It is a shareable choice to consider only well-differentiated carcinomas, but it should be stated in the methods section and it should be specified which type. Moreover, it should be examined also the outcome of these different neoplasms and their mortality rate.
Response:
We appreciate the reviewer for these helpful comments. Regarding the fact that more than 90% of thyroid cancer were differentiated thyroid cancer in Taiwan, our data were likely derived mainly from differentiated thyroid cancer. Additional studies are needed to evaluate the outcomes of patients with each histologic type of thyroid cancer as compared with that of general population. We have addressed this point in revised draft (lines 287-289, 307-310).

Round 2
Reviewer 3 Report
The authors have carefully investigated the critical issues highlighted by the reviewers, providing a reference on the effects of radioiodine therapy, while declaring that they have not assessed the association between the risk of second primary cancer mortality and this treatment. Therefore it was evidenced that further studies are needed both to evaluate the cardiovascular risk in patients with thyroid cancer and to define survival and death causes of thyroid cancer patients in other countries.
The authors admit that they did not differentiate the outcomes of the cases examined on the basis of the histological type of thyroid cancer, however they propose new studies in order to investigate the topic.
Overall, the paper results to be improved and deepened, English language and style are correct, the results obtained through the retrospective study are statistically significant and represented graphically by means of diagrams and tables and, for the innovative question to which it gives an answer, it deserves to be published.